

# Evaluating the potential of retinal photography in chronic kidney disease detection: a review

Nur Asyiqin Amir Hamzah[1,2], Wan Mimi Diyana Wan Zaki[1], Wan Haslina Wan Abdul Halim[3], Ruslinda Mustafar[3] and Assyareefah Hudaibah Saad[1]

[1] Faculty of Engineering and Built Environment, Universiti Kebangsaan Malaysia, UKM Bangi, Selangor, Malaysia
[2] Faculty of Engineering and Technology, Multimedia University, Ayer Keroh, Melaka, Malaysia
[3] Faculty of Medicine, Universiti Kebangsaan Malaysia, Cheras, Kuala Lumpur, Malaysia

Corresponding author
Wan Mimi Diyana Wan Zaki,
wmdiyana@ukm.edu.my

## ABSTRACT

**Background**. Chronic kidney disease (CKD) is a significant global health concern, emphasizing the necessity of early detection to facilitate prompt clinical intervention. Leveraging the unique ability of the retina to offer insights into systemic vascular health, it emerges as an interesting, non-invasive option for early CKD detection. Integrating this approach with existing invasive methods could provide a comprehensive understanding of patient health, enhancing diagnostic accuracy and treatment effectiveness.
**Objectives**. The purpose of this review is to critically assess the potential of retinal imaging to serve as a diagnostic tool for CKD detection based on retinal vascular changes. The review tracks the evolution from conventional manual evaluations to the latest state-of-the-art in deep learning.
**Survey Methodology**. A comprehensive examination of the literature was carried out, using targeted database searches and a three-step methodology for article evaluation: identification, screening, and inclusion based on Prisma guidelines. Priority was given to unique and new research concerning the detection of CKD with retinal imaging. A total of 70 publications from 457 that were initially discovered satisfied our inclusion criteria and were thus subjected to analysis. Out of the 70 studies included, 35 investigated the correlation between diabetic retinopathy and CKD, 23 centered on the detection of CKD *via* retinal imaging, and four attempted to automate the detection through the combination of artificial intelligence and retinal imaging.
**Results**. Significant retinal features such as arteriolar narrowing, venular widening, specific retinopathy markers (like microaneurysms, hemorrhages, and exudates), and changes in arteriovenous ratio (AVR) have shown strong correlations with CKD progression. We also found that the combination of deep learning with retinal imaging for CKD detection could provide a very promising pathway. Accordingly, leveraging retinal imaging through this technique is expected to enhance the precision and prognostic capacity of the CKD detection system, offering a non-invasive diagnostic alternative that could transform patient care practices.
**Conclusion**. In summary, retinal imaging holds high potential as a diagnostic tool for CKD because it is non-invasive, facilitates early detection through observable microvascular changes, offers predictive insights into renal health, and, when paired

> with deep learning algorithms, enhances the accuracy and effectiveness of CKD screening.

# INTRODUCTION

Chronic kidney disease (CKD) is a significant worldwide health problem with an estimated 11–13% of the population suffering from the disease (*Hill et al., 2016*). In 2017, nearly 700 million patients worldwide were affected which resulted in about 1.2 million fatalities due to complications associated with CKD (*Kassebaum et al., 2016*). Hence, efficient CKD detection tools are highly required considering these alarming statistics. The occurrence of CKD is determined by estimated Glomerular Filtration Rate (eGFR) of less than 60 mL/min per 1.73 m$^2$. The current practice of using blood and urine tests to measure albuminuria and eGFR primarily provides functional measures and their predictive precision is limited at earlier stages of the disease (*Sabanayagam et al., 2020*). Furthermore, albuminuria is very variable and can vary by up to 50% between individuals. Another method is examining kidney biopsy samples which is high predictive precision but is too invasive and expensive to use routinely (*Anderson et al., 1995*; *Hirsch & Brownlee, 2010*). Thus, a straight-forward and less invasive CKD detection is essential to help improve the screening procedure and prevent its progression and may encourage screening compliance among the at-risk community.

The retinal vasculature in the eyes provides a unique window into the systemic health of the body and may provide insightful information on the early stages of various diseases, including CKD. The eye and the kidney are closely related, sharing not only structural and developmental similarities but also genetic pathways. Numerous studies have identified associations between CKD and various ocular diseases, including age-related macular degeneration, diabetic retinopathy (DR), glaucoma, and cataracts (*Farrah et al., 2020*). In addition, there is a finding on dry eye disease can be an indicator of renal failure (*Trung et al., 2021*). Any changes in the retinal vessels could be signs associated with CKD, hypertension, diabetes, and cardiovascular disease (*London, Benhar & Schwartz, 2013*). Other common risk factors such as smoking, and obesity also contribute to both kidney and ocular diseases supporting the "Common Soil Hypothesis" (*Tham et al., 2020*). This suggests that the prevalence of visual impairment and ocular diseases is found to be significantly higher in persons with CKD compared to those without (*Wong et al., 2014*). Moreover, changes in retinal vascular parameters, including the diameter of the retinal artery and vein, fractal dimensions, tortuosity, and branch angle, have shown strong correlations with CKD progression (*Yip et al., 2017*; *Yau et al., 2011*; *Baumann, Burkhardt & Heemann, 2014*; *Grunwald et al., 2014*; *Bao et al., 2015*). Consequently, regular eye examinations in this population have become necessary in current medical procedures.

The non-invasive nature of digital fundus images has been critical in detecting DR (*Ramasamy et al., 2021*). This signifies the broader applicability of retinal imaging in medical diagnostics, supporting our motivation to explore its use in CKD detection. The current interpretation of the eye examination, however, is manual and can be further improved by employing an automated model. This requires the intervention of artificial intelligence (AI), particularly, machine learning to detect referable abnormalities in the retina (*Gulshan et al., 2016*). Deep learning, a subset of machine learning, which employs multi-layered neural networks, processes mass data to achieve high precision, and at times even surpasses human capabilities. The potential for increased time efficiency and reduced errors in manual interpretation makes deep learning applied to retinal imaging a promising area for researching kidney health assessment. The approach could not only complement the existing screening strategies but may encourage screening compliance among the at-risk community.

The use of retinal imaging for CKD detection is of particular interest to nephrologists, ophthalmologists, researchers focusing on diagnostic technologies, and AI developers working in healthcare. This approach not only offers a detailed view of changes in retinal vasculature associated with early stages of CKD but also serves as a less invasive alternative to kidney biopsies, which are costly and challenging to perform routinely. Scientifically, this review synthesizes current evidence linking retinal changes to CKD, advancing our understanding of the "Common Soil Hypothesis" which suggests shared pathophysiological pathways between the eye and the kidney. Clinically, integrating retinal imaging into the standard screening procedures for at-risk populations could enhance early detection rates, improve patient compliance with regular health assessments, and reduce the burden of CKD on healthcare systems.

This systematic literature review contributes significantly to the field in several aspects:

1. Tracing methodological evolution: Traces the evolution of methodologies in the detection of CKD, with a particular focus on techniques that analyze retinal vascular changes. This investigation offers a thorough understanding of how diagnostic approaches have progressed over time.
2. Retinal vascular analysis in diabetes: Investigates the progression of techniques utilizing retinal vascular analysis to identify kidney disease in individuals with diabetes. It covers the evolution of diagnostic approaches over time, with particular focus on the correlation between retinal changes and renal abnormalities in individuals with diabetes.
3. Role of retinal imaging in CKD detection: Highlights the evolving role of retinal imaging in CKD detection. This includes a focus on various methodologies that analyze retinal vascular changes based on retinal imaging.
4. Advancement of AI in CKD detection: Showcases the advantages of AI, particularly deep learning, in analyzing retinal images for detecting renal function and CKD. It highlights the evolution towards more accurate, non-invasive CKD screening methods, marking a significant advancement in the field.

Unlike existing reviews, our study provides a comprehensive analysis of the methodological evolution in retinal imaging for CKD detection, from manual techniques to

advanced deep learning approaches. We focus specifically on retinal vascular features and their correlations with CKD, integrating the latest advancements in AI to enhance diagnostic accuracy. By synthesizing novel evidence and discussing the clinical implications, our review offers unique insights that contribute significantly to the field. Through these contributions, this review aims to pave the way in CKD detection, focusing on the capabilities of image processing techniques to extract significant features from retinal imaging based on vascular changes, and on the potential for utilizing AI to enhance diagnostic accuracy and efficiency.

This paper is structured into four sections. The Survey Methodology section outlines the systematic approach used for article selection and data analysis, followed by reviews of the progressions in retinal imaging for CKD detection. The Challenges and Way Forward section discusses current limitations and future directions in this research area. Finally, the paper concludes with a summary of findings and recommendations for future research in CKD detection using retinal imaging and AI.

## SURVEY METHODOLOGY

An extensive review of literature on a specific topic was conducted to examine and assess the development of different strategies for CKD detection based on retinal vascular changes. This research aims to organize and comprehend the scrutinized publications from leading academic search platforms, including Google Scholar and chosen publisher databases, with citation indices tracked by SCImago journal rankings. These include, but are not limited to, ACM Digital Library, Cochrane, Engineering Journal, IEEE Xplore, PubMed, Science Direct, Scopus, SpringerLink and World of Science.

A combination of text words and medical subject headings will be used where necessary. The search strategy incorporated a combination of medical subject headings and free-text words. Search terms included "kidney", "renal", "retina", "eye", "ocular", and "macular", combined with "fundus" using Boolean operators. For instance, the combinations used were: ("kidney" OR "renal") AND ("retina" OR "eye" OR "ocular" OR "macular") AND "fundus". The search criteria were deliberately broad and intentionally did not specify stages of CKD or techniques due to the use of general search terms. This was done to ensure a comprehensive overview of the available literature on the topic.

The search initially resulted in the identification of 457 articles, which then underwent a detailed refinement process to narrow down the selection to those most relevant to the study's objectives. This process began with the removal of 21 duplicate articles that were identified, thus ensuring no repetition in the resources considered. Following this, a review of the titles led to the exclusion of 314 articles. The basis for this substantial reduction was the determination that these articles did not specifically align with the review's focus, particularly their relevance to the detection of renal failure through retinal changes. The refinement process proceeded with the remaining 122 abstracts, which were subjected to a rigorous evaluation. This evaluation was guided by clearly defined exclusion criteria, emphasizing the relevance to the study's objectives, the necessity for empirical data supporting CKD detection through retinal changes, and the overall quality of the study. This meticulous examination resulted in the further exclusion of 52 articles.
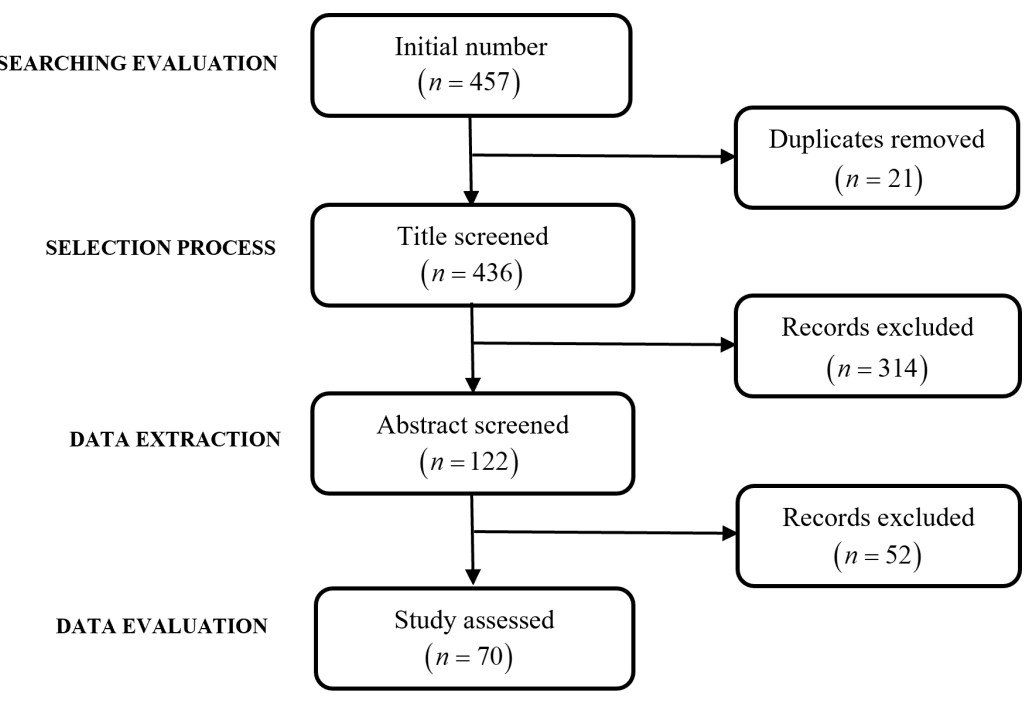

Where $n$ is the number of articles

**Figure 1** Flowchart of records identification and selection.

The criteria for excluding articles were specific and methodically applied to ensure the relevance and quality of the selected studies. Exclusion criteria were applied rigorously:

- Articles were excluded if they only peripherally mentioned retinal changes without direct correlation to CKD detection.
- Reviews and theoretical papers without new empirical findings were omitted.
- Studies lacking clear methodological descriptions or that did not focus on CKD stages detectable through retinal imaging were also excluded.
- Articles not available in full text.

The article selection process was facilitated by Mendeley, a citation management software, ensuring organized and efficient handling of the literature. The flowchart in Fig. 1 offers a visual representation of this systematic process, from initial identification to final article selection process for this review based on PRISMA guidelines (*Abdolrasol et al., 2021*).

## EVOLUTION OF RETINAL VASCULAR ANALYSIS IN CHRONIC KIDNEY DISEASE DETECTION

Chronic kidney disease (CKD) has been linked to a range of systemic complications, including diabetic retinopathy, a common complication of diabetes that affects the retina. Several studies suggest a correlation between the presence and severity of diabetic

retinopathy and CKD in patients with diabetes, sparking interest in the investigation of retinal alterations specific to CKD. Advancements in retinal imaging techniques, such as digital fundus image and optical coherence tomography (OCT), have facilitated detailed retinal analysis, enabling researchers to pinpoint potential CKD indicators. Furthermore, certain retinal microvascular abnormalities, like arteriovenous nicking and retinopathy, have been linked to an elevated CKD risk. The surge in AI and machine learning technologies has propelled efforts to automate retinal image analysis for CKD prediction or detection. By training algorithms on extensive retinal image datasets from patients with confirmed CKD stages, researchers are aspired to identify early CKD signs solely from retinal images. Despite the promise of this approach, the utilization of retinal changes for CKD detection remains experimental, necessitating broader studies, validation, and technological refinement to transit it into a mainstream diagnostic instrument. Hence, the following section reviews studies related to the relationship between DR and CKD, the role of retinal imaging, the significance of retinal microvascular abnormalities, and the potential of AI and machine learning.

## Diabetic retinopathy and CKD

Understanding the connection between kidney and ocular diseases can lead to the development of innovative treatment and screening strategies for both conditions, ultimately improving the quality of life for affected individuals. The complex relationship between the eyes and kidneys is particularly evident in individuals with diabetes. DR is a common microvascular complication of diabetes that has been found to have a compelling association with CKD in diabetic patients (*Bermejo et al., 2023*; *Lee, Lee & Kim, 2023*). The existence and severity of DR often mirror the presence and stage of CKD, suggesting that the eye can serve as a window to systemic health, particularly kidney function. Figure 2 shows a side-by-side comparison of a healthy retina and one affected by DR, highlighting the pathological changes that occur in the retinal vasculature. These changes are visible through retinal imaging, correlate with similar microvascular damage in the kidneys seen in CKD, highlighting the potential of ocular assessments in early CKD detection and monitoring.

This section explores the association between DR and CKD, examining the underlying pathophysiological mechanisms and assessing the potential utility of retinal changes as early indicators for CKD in diabetic patients.

### *Diabetic retinopathy as a marker for CKD*

The exploration of DR as a potential marker for CKD has been a focal point in several studies, revealing a complex relationship. *Yip et al. (2015)* conducted a study that highlighted the association between retinopathy and both prevalent and incident end-stage renal disease (ESRD) in a multi-ethnic Asian population. Their findings suggested that retinopathy could serve as an early indicator of subclinical damage in the renal microvasculature, particularly among diabetic individuals. Similarly, *Park et al. (2019)* examined into the relationship between the severity of DR and the progression of CKD in patients with type 2 diabetes mellitus (T2DM). Their research indicated that the severity of

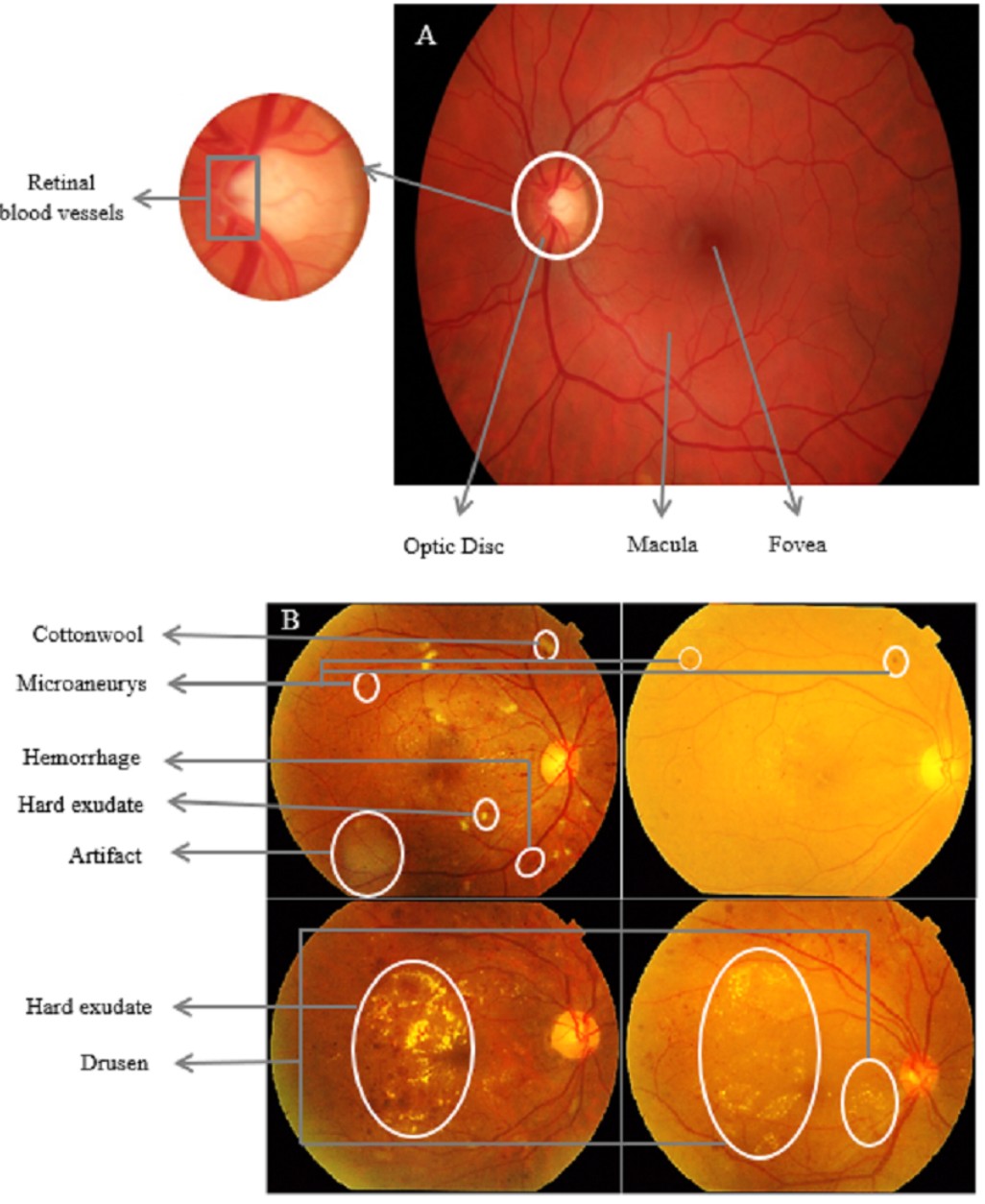

**Figure 2 Comparative retinal imaging in health and diabetic retinopathy.** (A) A normal retina with clear macula and well-defined retinal blood vessels. (B) DR characterized by microaneurysms, cotton wool spots, and neovascularization. Such retinal abnormalities, particularly when quantified using retinal imaging techniques, may provide insights into the microvascular complications associated with chronic kidney disease (*Zaki, Hussain & Mutalib, 2018*).

DR, categorized into non-proliferative DR (NPDR) and proliferative DR (PDR), could act as a prognostic indicator for CKD progression. The study emphasized the importance of early DR severity evaluation and rigorous monitoring of renal function and albuminuria in patients with significant DR.

In addition to these studies, other research has also explored the link between retinal changes and renal function. For instance, *Bao et al. (2015)* examined the association of retinal vessel diameter with CKD in a rural Chinese cohort, observing that narrower retinal arterioles were linked to increased albuminuria, a marker for kidney damage. However, their study did not demonstrate a significant relationship between the presence of retinopathy and the incidence of CKD or albuminuria across the general population. *Hwang et al. (2016)* found that CKD patients with retinopathy had a heightened prevalence of vascular calcification in the abdominal aorta and iliofemoral artery, suggesting a connection between retinal and renal health.

Moreover, studies like those conducted by *Zhao et al. (2021)* and *Nusinovici et al. (2021)* further focused the quantitative relationships between retinal vascular characteristics and clinical indicators of renal function. Their findings disclose significant correlations, such as between DR severity and renal indicators like albumin to creatinine ratio, as well as between wider retinal vascular calibers and heightened risk of diabetic kidney disease, which could be instrumental for early detection and management of diabetes-related kidney complications. Collectively, these studies underline the potential of DR as an early marker for renal dysfunction in diabetic patients. They highlight the need for vigilant monitoring of retinal health to predict and potentially mitigate the progression of CKD in diabetic populations.

### *The contrasting findings*

While numerous studies have highlighted a significant association between retinal changes and CKD, some research presents contrasting findings, highlighting the complexity of this relationship. Although *Bao et al. (2015)* found an association between retinal arteriovenous ratio (AVR) and albuminuria, suggesting a possible vascular component in renal impairment, however, the absence of a significant connection with CKD and retinopathy in a broader cohort indicates a complex connection rather than a straightforward relationship.

*McKay et al. (2018)* conducted a study that found retinal microvascular parameters, such as vascular spread, tortuosity, and branching patterns, were not significantly associated with reduced renal function in individuals with type 2 diabetes. Their findings suggested that retinal microvascular parameters did not forecast eGFR degradation over an average of 3 years, challenging the idea of a direct and consistent link between retinal and renal health. Similarly, *Keel et al. (2017)* aimed to evaluate the relationship between retinal vascular caliber and kidney function in a cohort of Australian children and adolescents with type I diabetes. Their findings indicated that retinal vascular caliber was not significantly associated with microalbuminuria in the sample, adding another layer of complexity to the understanding of the relationship between retinal and renal health.

Meanwhile *Wang et al. (2023)* provides a more disease-specific insight, linking lower eGFR levels to an increased risk of DR while dissociating eGFR and microalbuminuria

levels from diabetic macular edema. This compilation of studies highlights the potential of retinal imaging in indicating systemic vascular health, but also cautions against a one-size-fits-all approach, emphasizing the need for further investigation in the relationship of DR with potential renal complications. These studies emphasize the complex nature of the relationship between retinal changes and CKD. They suggest that while there is a connection, it may be influenced by various factors such as age, diabetes type, and the specific retinal parameters studied, and may not be universally applicable across different populations and conditions.

### The promising potential of retinal microvasculature changes in diabetic retinopathy for early detection of CKD

The exploration of retinal microvasculature changes in DR as a predictive tool for CKD has shown promising results. Studies such as those conducted by *Wang et al. (2020)* and *Zhuang et al. (2020)* have employed advanced imaging techniques like optical coherence tomography angiography (OCTA) to demonstrate a significant association between retinal vessel density and renal function in patients with type 2 diabetes. Similarly, *Yan et al. (2023)* observed a higher prevalence of DR in patients with diabetic nephropathy, pointing out the potential relationship between retinal changes and kidney function. The predictive potential of retinal evaluations is further highlighted by research from *Wang et al. (2023)*, which found that low eGFR levels and increasing microalbuminuria levels corresponded with the progression of DR. The integration of AI has also shown promise in enhancing early detection capabilities. For instance, *Zhang et al. (2021)* developed an AI system that could diagnose CKD and predict its onset using retinal fundus images.

However, it is important to acknowledge certain complexities and challenges, as highlighted by previous works *Bao et al. (2015)*, *Keel et al. (2017)*, *McKay et al. (2018)* and *Wang et al. (2023)*. These studies concluded that the relationship between retinal microvascular parameters and renal function was not always straightforward, suggesting that various factors may influence this connection (*Bao et al., 2015*; *Keel et al., 2017*; *McKay et al., 2018*; *Wang et al., 2023*). In conclusion, while there are challenges to be addressed, the collective findings from these studies suggest that the potential of utilizing retinal microvasculature changes in DR as a means of detecting CKD is promising. These findings promote further exploration and validation to fully harness retinal assessments in the early detection and management of CKD in diabetic patients.

To provide a concise overview of the findings and relationships identified in the literature, Table 1 summarizes key aspects of the studies, including the retinal vasculature features observed, the kidney pathologies identified, and the established relationships between these two.

## Retinal imaging & retinal microvascular abnormalities

The detailed analysis of retinal vasculature offers significant promise in the early detection and monitoring of CKD. Over the years, advancements in retinal imaging techniques, such as fundus photography and OCTA, have equipped researchers and clinicians with powerful tools to visualize and assess the retinal vasculature with unprecedented detail. These techniques facilitate the identification of potential markers that might indicate the

**Table 1** Relationship between retinal pathologies/features and kidney pathologies/biomarkers in diabetic patients.

| Authors (Year) | Retinal imaging modality/technique | Retinal pathologies/features | Kidney pathologies/biomarkers | Direct correlation? | | Relationship between retinal and kidney |
|---|---|---|---|---|---|---|
| | | | | Yes | No | |
| *Schwartz et al. (1998)* | – | DR | Kimmelstiel-Wilson nodules; Mesangial sclerosis lesions; Non-nodular diabetic; Glomerulosclerosis | ✓ | | Severe kidney lesions correlate with worse retinopathy, mesangial sclerosis is linked to mild diabetic retinopathy, and the absence of nodular glomerulosclerosis indicates no proliferative retinopathy. |
| *Yazdani et al. (1995)* | Retinal photography, FFA | Microangiopathic changes; Arterial changes; Delayed choroidal filling on fluorescein angiography; Exudates and hemorrhages | Microvascular changes in diabetic nephropathy; Chronic renal failure on hemodialysis; Proteinuria | ✓ | | Diabetic retinopathy and nephropathy exhibit diabetes-induced damage in eyes and kidneys; early eye vessel damage corresponds with kidney protein leaks, and severe eye damage in diabetes reflects chronic kidney failure. |
| *Wong et al. (2004a)* | Retinal photography | Retinal hemorrhages; Microaneurysms; Soft exudates; Arteriovenous nicking | Renal dysfunction | ✓ | | Retinopathy and kidney dysfunction both indicate systemic microvascular damage, with microaneurysms, hemorrhages, and arteriovenous nicking in the eye reflecting similar kidney damage. |
| *Wong et al. (2004b)* | Retinal photography | Retinal vessels diameter (arteriolar and venular) | Gross proteinuria; Renal insufficiency | ✓ | | Enlarged retinal venular diameter indicates kidney damage. |
| *Klein et al. (2007)* | Retinal photography | Retinal arterioles average caliber; Retinal venules average caliber | Nephropathy (Incident) | ✓ | | Retinal vessel caliber is independently associated with the nephropathy. |
| *Grauslund et al. (2009)* | Retinal photography | CRAE; CRVE | Nephropathy | ✓ | | Narrower arterioles is associated with nephropathy. |
| *Klein et al. (2010)* | Retinal photography | CRAE; CRVE | Diabetic nephropathy lesion development; Glomerulopathy index (in 5 years); Mesangial matrix fractional volume; Volume fraction of cortex that was interstitium. (in 5 years) | ✓ | | CRAE is associated with changes in the glomerulopathy index over 5 years; a reduction in arteriole diameter indicates an increase in mesangial matrix accumulation in the kidneys, and CRVE is related to changes in the volume fraction of the renal cortex. |

**Table 1** (*continued*)

| Authors (Year) | Retinal imaging modality/technique | Retinal pathologies/features | Kidney pathologies/biomarkers | Direct correlation? | | Relationship between retinal and kidney |
|---|---|---|---|---|---|---|
| | | | | Yes | No | |
| *Pedro et al. (2010)* | Retinal photography | DR; Diabetic Macular Edema | MAU; Overt Nephropathy | ✓ | | MAU is a risk factor for type 1 diabetic retinopathy, and overt nephropathy is correlated with diabetic retinopathy. |
| *Gao et al. (2011)* | Retinal photography | Retinopathy; Glaucoma suspect; AMD | CKD | ✓ | | CKD patients have a higher prevalence of retinopathy, increased rates of glaucoma suspect and AMD, and a significant association between proteinuria and retinal complications/overall eye pathology. |
| *Deva et al. (2011)* | Retinal photography | Incidental retinal abnormalities; Microvascular retinopathy; DR; Macular degeneration | CKD stages 3 to 5; Renal failure | ✓ | | CKD patients (stages 3–5) exhibit a higher rate of retinal abnormalities, with advanced CKD or renal failure patients commonly having microvascular retinopathy, diabetic retinopathy, and macular degeneration issues. |
| *Grunwald et al. (2012)* | Retinal photography | Retinopathy; Vascular abnormalities; Vessel diameter caliber | CKD | ✓ | | Greater severity of retinopathy indicates higher CVD prevalence, with retinal vascular abnormalities related to hypertension, and CVD prevalence directly correlates with mean venular caliber in the retina. |
| *Sasongko et al. (2012)* | Retinal photography | Retinal arteriolar tortuosity; DR | Diabetic nephropathy (early stage) | ✓ | | Increased tortuosity in retinal arterioles indicates both retinal and early kidney dysfunction. |
| *Benitez-Aguirre et al. (2012)* | Retinal photography | Length-to-diameter ratio; Simple tortuosity | Renal dysfunction (AER) | ✓ | | Higher venular LDR and lower venular ST in retinal vessels predict renal dysfunction, with retinal vascular changes correlating with renal dysfunction and increased AER. |

**Table 1** (*continued*)

| Authors (Year) | Retinal imaging modality/technique | Retinal pathologies/features | Kidney pathologies/biomarkers | Direct correlation? | | Relationship between retinal and kidney |
|---|---|---|---|---|---|---|
| | | | | **Yes** | **No** | |
| *Liew et al. (2013)* | Retinal photography | Retinopathy lesions; Retinal venules (venular dilation) | CKD | ✓ | | Retinopathy lesions and wider retinal venules are linked to CKD in both diabetic and non-diabetic individuals. |
| *Nagaoka & Yoshida (2013)* | Laser doppler velocimetry system | Retinal blood flow; Vessel diameter | CKD | ✓ | | Stage 3 CKD patients showed a significant decrease in RBF due to reduced vessel diameter, without changes in blood velocity. |
| *Zhang et al. (2014)* | Retinal photography | NPDR; PDR | eGFR; CKD; MAU | ✓ | | The severity of DR (NPDR to PDR) correlates with lower eGFR, with more severe DR linked to higher retinol-binding protein levels and an increased albumin/creatinine ratio (ACR). |
| *Baumann, Burkhardt & Heemann (2014)* | Retinal photography | Retinal arteriolar | MAU; CKD (stage 2–4) | ✓ | | Retinal arteriolar narrowing is significantly associated with renal end points in CKD. |
| *Grunwald et al. (2014)* | Retinal photography | Retinopathy; Vessel calibers | CKD; ESRD; eGFR | ✓ | | The link between retinopathy and kidney disease progression weakened, indicating an influence from baseline kidney health, and there is a complex relationship between retinal arteriole/vein ratio and ESRD risk. |
| *Yip et al. (2015)* | Retinal photography | Retinopathy; Retinal microvascular signs | ESRD | ✓ | | There is significant link between retinopathy and both existing and new cases of ESRD, particularly in diabetic individuals, with alterations in retinal microvascular signs mirroring kidney microvascular damage; however, retinopathy might not greatly raise the risk of ESRD in non-diabetics, indicating a weaker link between eye and kidney diseases in this group. |

**Table 1** (*continued*)

| Authors (Year) | Retinal imaging modality/technique | Retinal pathologies/features | Kidney pathologies/biomarkers | Direct correlation? | | Relationship between retinal and kidney |
|---|---|---|---|---|---|---|
| | | | | **Yes** | **No** | |
| *Bao et al. (2015)* | Retinal photography | AVR | CKD; Albuminuria; eGFR | ✓ | | Lower retina AVR is associated with albuminuria. |
| | Retinal Photography | Retinopathy | CKD; Albuminuria; eGFR | | ✓ | No significant link between retinopathy and CKD or albuminuria in the general population. |
| *Hwang et al. (2016)* | Retinal photography | Retinopathy | CKD (stages 3–5); VCL | ✓ | | Patients with retinopathy had more VCL in key arteries, and both conditions were associated with accelerated kidney function decline. |
| *Keel et al. (2017)* | Retinal photography | Retinal vascular caliber | MAU | | ✓ | No significant link between retinal vascular caliber and MAU was found. |
| *McKay et al. (2018)* | Retinal photography | Microvascular parameters (vascular spread, tortuosity, branching patterns) | eGFR | | ✓ | No significant link between microvascular parameters and eGFR, and the initial association between retinal arteriolar diameter and eGFR was not sustained after adjustments. |
| *Vadalà et al. (2019)* | OCT/OCTA | Choroidal thickness | Albuminuria; Glycated Hemoglobin (HbA1c) Levels | ✓ | | Significant positive correlation found between choroidal thickness and both albuminuria and HbA1c levels. |
| *Park et al. (2019)* | Retinal photography | DR; NPDR; PDR | CKD; Renal function and albuminuria | ✓ | | Increased severity of DR correlates with higher CKD progression risk, with NPDR and PDR patients having a 2.9-fold and 16.6-fold higher risk, respectively. |
| *Wang et al. (2020)* | OCT/OCTA | Retinal vessel density | eGFR; MAU | ✓ | | Lower eGFR and higher MAU levels correlate with reduced retinal vessel density. |

**Table 1** (*continued*)

| Authors (Year) | Retinal imaging modality/technique | Retinal pathologies/features | Kidney pathologies/biomarkers | Direct correlation? | | Relationship between retinal and kidney |
|---|---|---|---|---|---|---|
| | | | | **Yes** | **No** | |
| *Zhuang et al. (2020)* | OCT/OCTA | Retinal vessel density; Macular thickness | eGFR; Urinary albumin-to-creatinine ratio; Hemoglobin/Hematocrit | ✓ | | Lower eGFR is linked to reduced SVC vessel density in the macula, higher UACR is associated with increased macular thickness, and reduced HGB/HCT correlates with both decreased SVC vessel density and increased macular thickness. |
| *Xu et al. (2020)* | Retinal photography | Peripheral vascular calibers; Arteriolar geometries | MAU | ✓ | | A strong link was found between retinal blood vessel width changes and early kidney damage, with a significant association between arteriolar geometries and MAU, indicating that retinal arteriolar structure changes may signal renal dysfunction. |
| *Zhang et al. (2021)* | Retinal photography | Retinal fundus abnormalities detected by AI | CKD; T2DM | ✓ | | The AI system detects CKD by identifying specific retinal features in fundus images, suggesting retinal alterations as indicators of CKD. |
| *Zhao et al. (2021)* | Retinal photography | DR; Df | Albumin to creatinine ratio; Uric acid; creatinine; Albumin; eGFR | ✓ | | A significant correlation was found between DR severity and key renal indicators (albumin to creatinine ratio, uric acid, creatinine, albumin, eGFR), suggesting worsening kidney function mirrors retinal damage in diabetes, and a higher fractal dimension in retinal vessels strongly correlates with renal function parameters, indicating that more complex retinal vessel patterns may reflect deteriorating kidney health. |

**Table 1** (*continued*)

| Authors (Year) | Retinal imaging modality/technique | Retinal pathologies/features | Kidney pathologies/biomarkers | Direct correlation? | | Relationship between retinal and kidney |
|---|---|---|---|---|---|---|
| | | | | **Yes** | **No** | |
| *Nusinovici et al. (2021)* | Retinal photography | CRAE; CRVE; DR | Diabetic Kidney Disease (DKD) | ✓ | | Wider CRAE and CRVE are linked to an increased risk of DKD. |
| *Zeng et al. (2021)* | OCT/OCTA | Retinal ganglion cell-inner plexiform layer thickness; Ganglion cell complex-focal loss volume & ganglion cell complex-global loss volume; Retinal vessel density in superficial vascular plexus | CKD | ✓ | | A decline in GCIPLT across all CKD stages indicates increased retinal neurovascular impairment with worsening kidney function; elevated focal and global loss volumes in the ganglion cell complex in CKD patients imply retinal ganglion cell damage linked to kidney function decline, and lower retinal vessel density in advanced CKD stages (3–5) compared to early stages (1–2) suggests reduced vascular integrity in the retina with severe CKD. |
| *Iwase et al. (2023)* | LSFG | Retinal blood flow (MBR); Retinal microcirculation; Vascular parameters | CKD | ✓ | | Stage 3 CKD patients had lower mean blur rate and total retinal flow index, indicating reduced retinal blood flow and suggesting worsening CKD affects retinal circulation; diabetic patients with CKD show altered retinal microcirculation, highlighting a complex relationship between diabetes, CKD, and retinal health. |
| *Yan et al. (2023)* | Retinal photography, OCT/OCTA, FFA | DR; Hard Exudates; DME | Diabetic nephropathy; Urine ACR stage | ✓ | | ACR Stage & DR: Significant correlation found between ACR stages (reflecting DN severity) and DR presence |

**Table 1** (*continued*)

| Authors (Year) | Retinal imaging modality/technique | Retinal pathologies/features | Kidney pathologies/biomarkers | Direct correlation? | | Relationship between retinal and kidney |
|---|---|---|---|---|---|---|
| | | | | **Yes** | **No** | |
| | Retinal photography, OCT/OCTA, FFA | DR | eGFR; MAU | ✓ | | Lower eGFR levels are linked to an increased risk of developing DR, and higher microalbuminuria levels correlate with the progression of DR to more severe stages. |
| *Wang et al. (2023)* | Retinal photography, OCT/OCTA, FFA | DME | eGFR; MAU | | ✓ | No significant relationship found between eGFR/MAU levels and the development of DME, suggesting renal damage's impact on DME is unclear. |

**Notes.**

AER, Albumin excretion rate; DR, Diabetic retinopathy; AMD, Age-related macular degeneration; eGFR, estimated Glomerular Filtration Rate; AVR, Arteriole-to-venule ratio; ESRD, End-stage renal disease; CKD, Chronic kidney disease; FFA, Fluorescence angiography; CRAE, Central Retinal Arteriolar Equivalent; GCIPL, Ganglion cell-inner plexiform layer; CRVE, Central Retinal Venular Equivalent; LSFG, Laser Speckle Flowgraphy; CVD, Cardiovascular disease; MAU, microalbuminuria; Df, Fractal dimension; NPDR, Non-proliferative diabetic retinopathy; DKD, Diabetic Kidney Disease; PDR, Proliferative diabetic retinopathy; DME, Diabetic macular edema; VCL, Vascular calcification.

presence or risk of developing CKD. This section investigates modern retinal imaging techniques, emphasizing how they have revolutionized the capacity to analyze the retina and potentially detect CKD at an early stage, hence enabling timely intervention.

### Retinal vasculature

Micro-vessels, with a luminal diameter of less than 300 micrometers, are crucial in controlling tissue blood flow and influencing systemic vascular resistance, a function intrinsically related to endothelial performance. Multiple pathological events can both induce and result from endothelial malfunction, subsequently affecting the micro-vessels (*Farrah et al., 2020*). The inception and implications of microvascular disease include a range of pathological changes and risk factors. For instance, endothelial dysfunction is associated with a variety of risk factors including decreased kidney function, high blood pressure, smoking, elevated levels of glucose and insulin, and dyslipidemia. Meanwhile, micro-vessel dysfunction is characterized by a series of pathological alterations including decreased fibrinolysis, reduced ability to respond to vasodilators, heightened platelet reactivity, vascular remodeling, and rarefaction, all of which collectively contribute to an increase in vascular resistance and a reduction in blood supply.

Modifications in microvascular structure and function are significant factors in the initiation and progression of hypertension, diabetes, CKD, and cardiovascular disease (CVD) (*Deanfield, Halcox & Rabelink, 2007*; *Houben, Martens & Stehouwer, 2017*; *Stehouwer, 2018*). Remarkably, these alterations manifest before any perceptible damage to the end organs and seem to be correctable (*Halcox et al., 2002*; *Remuzzi et al., 2016*). Additionally, microvascular impairments in peripheral tissues reflect similar malfunctions

in visceral organs (*Anderson et al., 1995*; *Bonetti et al., 2004*), justifying the examination of easily accessible micro-vessels, like those in the eye. The clarity of the eye's media enables direct observation of the microvasculature, which can be compromised by systemic conditions such as hypertension, diabetes, and CKD.

The relationship between retinal vascular alterations and CKD has been a subject of multiple studies. Although there has been investigation into the connection between the caliber of retinal vessels and the onset of CKD, findings remain inconclusive. For instance, a study conducted by *Yau et al. (2011)* did not find any correlation between retinal microvascular caliber and the onset of stage 3 CKD in the general populace; however, it did find that among white individuals, narrower arterioles were correlated with an increased likelihood of developing stage 3 CKD. Similarly, research by *Sabanayagam et al. (2011)* did not establish any significant correlation between the diameter of retinal vessels and the likelihood of reduced kidney function. As such, both studies implied that although there might be shared mechanisms between retinal vessel diameters and CKD, they are not directly causally related. In a nutshell, while alterations in retinal vasculature may signal CKD in specific demographic groups, further studies are necessary to comprehensively understand the connection between changes in retinal vasculature and CKD.

### Retinal imaging techniques and CKD detection

The exploration of retinal imaging as a diagnostic tool for CKD has attracted significant attention in recent research, revealing the promising potential of detailed analysis of retinal vasculature in the early detection and monitoring of CKD. Progressions in retinal imaging techniques, such as fundus photography and OCT, have equipped researchers and clinicians with powerful tools to visualize and assess the retinal vasculature with exceptional detail. These techniques facilitate the identification of potential markers that might indicate the presence or risk of developing CKD. A multitude of studies have explored the complex relationship between retinal and renal health, leveraging these advanced imaging techniques alongside machine learning algorithms.

Innovations in retinal imaging, particularly fundus imaging and OCTA, have been crucial in advancing our understanding of the link between retinal microvascular changes and CKD. As presented in Table 1, *Zhuang et al. (2020)* demonstrated that OCTA could detect preclinical microvascular impairments in DR, which are indicative of CKD. The study highlighted OCTA's ability to measure retinal vessel density and thickness, revealing a significant correlation between reduced vessel density in the superficial vascular complex and impaired renal function. In addition, *Xu et al. (2020)* expanded the scope of retinal vascular analysis using digital fundus photography to include peripheral vascular calibers and arteriolar geometries, finding strong associations with microalbuminuria, an early marker of renal dysfunction in type 2 diabetes. *Kang et al. (2020)* leveraged deep learning models with retinal fundus imaging to detect early renal function impairment, achieving area under curve (AUC) of 0.81, which highlights the potential of retinal imaging in identifying systemic cardiovascular risks. *Zhao et al. (2021)* developed a deep learning methodology for retinal vessel segmentation, which, when combined with OCTA, revealed

significant correlations between the intricacies of retinal vessel structure, such as fractal dimension, and various markers of renal function across various stages of DR.

Meanwhile, *Frost et al. (2021)* used OCTA to observe retinal capillary rarefaction, finding it to be associated with reduced eGFR in hypertensive patients, suggesting that retinal microvascular health reflects systemic vascular integrity. *Paterson et al. (2021)* employed the Vessel Assessment and Measurement Platform for Images of the Retina (VAMPIRE) software to quantify retinal microvascular parameters, discovering a strong link between reduced fractal dimension and increased odds of albuminuria, independent of diabetes and blood pressure. *Peng et al. (2021)* identified a correlation between retinal deep vascular plexus vessel density and early cognitive deficiencies in CKD patients, suggesting retinal neurovascular biomarkers as potential indicators of cognitive impairment. *Yong et al. (2022)* measured OCTA metrics across different CKD retinal characteristics and found significant differences in vascular density and perfusion density, which varied based on the underlying cause of CKD. *Yan et al. (2023)* documented a substantial relationship between diabetic nephropathy and retinal microvascular changes, such as DR and hard exudates, using a combination of color fundus imaging, OCT, and fluorescence angiography.

Additionally, *Wang et al. (2023)* conducted a three-year study using fundus photography and swept-source OCT to explore the relationship between renal function and DR, finding that lower eGFR and higher microalbuminuria levels were associated with the progression of DR, although no direct link to diabetic macular edema was established. These studies collectively enlighten the critical role of retinal imaging in detecting and monitoring CKD, offering a window into the systemic microvascular changes associated with the disease. An interesting technique has been employed by *Mustafar et al. (2023)* where they combined fundus photography for assessing retinal changes, OCT for macular volume measurements, and blood sample analyses for cardiac biomarkers, when examining the relationship between retinal vessel changes and cardiac biomarker levels in patients with various stages of CKD. The findings suggest a negative correlation between eGFR and retinal vessel tortuosity, a positive correlation between proteinuria and central retinal venular equivalent (CRVE), and various associations with cardiac biomarkers like high-sensitivity C-reactive protein. These findings indicate the potential of using retinal and cardiac biomarkers as non-invasive tools for assessing CKD.

Figure 3 illustrates the application of retinal imaging—fundus photography and OCT, as investigative tools for the detection of CKD, showcasing their potential utility in identifying biomarkers associated with renal pathology.

### Promising potential of fundus imaging in early detection of CKD

The exploration of fundus images as a diagnostic tool for CKD has shown promising results. Fundus imaging, being a non-invasive technique, offers a unique advantage in facilitating early detection of CKD. A significant aspect of this research is the established correlation between retinal pathologies and key kidney function markers. Studies by *Edwards et al. (2005)*, *Baumann et al. (2010)*, and *Sabanayagam et al. (2009)* have demonstrated a direct association between conditions like retinopathy and arteriolar abnormalities with serum creatinine levels and eGFR. This suggests that retinal changes can serve as early

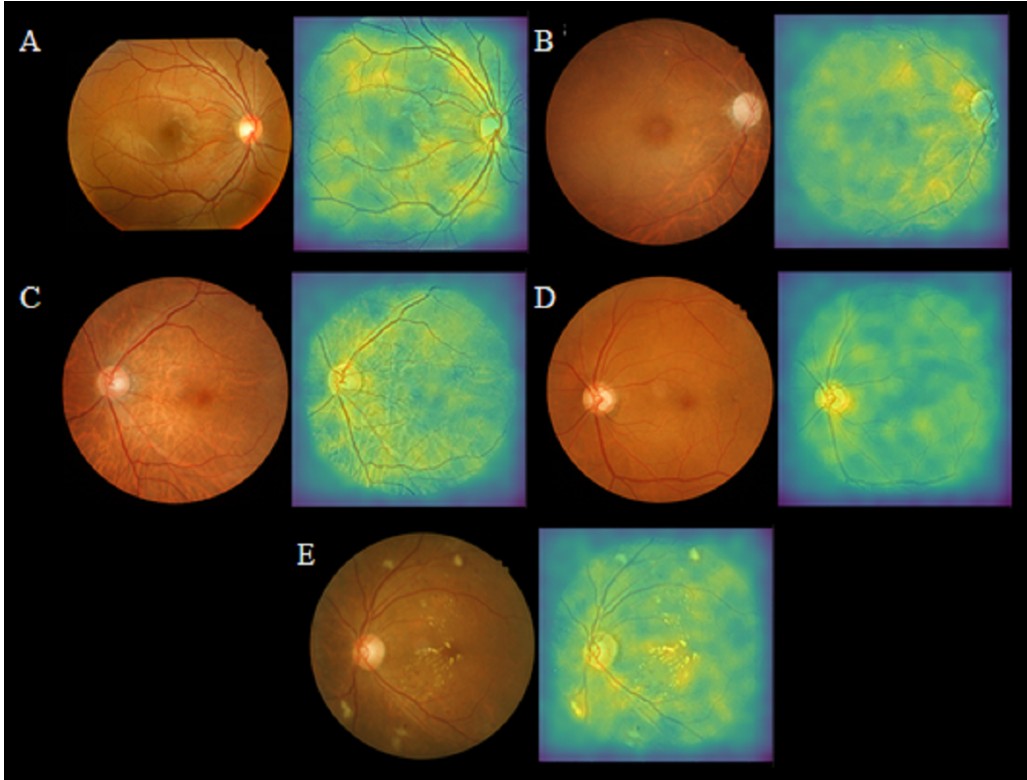

**Figure 3** **Selected retinal fundus images and their corresponding saliency maps in true-negative and true-positive cases, adapted from work done by *Kang et al. (2020)* in the attempt to detect of early renal function impairment using fundus images.** (A) No renal function impairment detected. (B) Renal function impairment detected. (C) Renal function impairment detected. (D) Renal function impairment detected (E) Renal function impairment detected.

indicators of kidney health. Furthermore, the impact of vascular changes in the retina, such as arteriolar narrowing, has been linked to microvascular damage associated with CKD. This is supported by the research of *Yau et al. (2011)* and *Ooi et al. (2011)* and is further evidenced in the work of *Grunwald et al. (2010)*. In addition, *Kang et al. (2020)*, who attempted to automate the detection achieved an impressive area under curve (AUC) of 0.81 in detecting renal function impairment using deep learning models applied to fundus images.

*Peng et al. (2020)*, *Peng et al. (2021)* reveal a complex relationship between retinal vascular health and kidney function in CKD. Their previous work *Peng et al. (2020)* found that blood pressure variability in CKD patients negatively correlates with retinal vessel density, suggesting that retinal microvascular integrity may reflect broader vascular health. However, the latter showed no significant link between retinal biomarkers and cognitive impairment in CKD, indicating that while retinal changes mirror microvascular damage, they do not necessarily predict cognitive decline in these patients. This highlights the potential yet complicated role of retinal imaging as an indicator of systemic vascular conditions.

The versatility of fundus imaging in adapting to various imaging techniques and algorithms for tailored retinal vessel segmentation, as demonstrated in the work of *Zhao et al. (2021)* and *Zhuang et al. (2020)*, further emphasizes its compatibility and potential for widespread clinical application. The evidence in Table 2 clearly shows that retinal imaging is becoming important for detecting and managing CKD early. This approach, particularly when combined with advanced analytical methods like deep learning, holds significant promise. However, the need for continued research and validation is critical to fully harness and refine these diagnostic capabilities in the clinical setting.

## AI and retinal microvasculature for potential CKD detection

Through the deployment of advanced algorithms, retinal images are now being studied with unparalleled precision. By exploiting the robustness of machine learning and training these algorithms on extensive datasets of retinal images from patients with delineated CKD stages, the potential for detecting early CKD signs from retinal images alone has surged. These AI-driven methodologies are not merely supplementing the capabilities of conventional imaging, but they are also setting the stage for cutting-edge advancements. The research was conducted by *Sabanayagam et al. (2020)*, *Kang et al. (2020)*, *Zhang et al. (2021)* and *Zhao et al. (2021)* emphasize the applications of AI in retinal imaging for CKD detection. These studies utilize deep learning algorithms on extensive datasets of retinal images, demonstrating the feasibility of non-invasive screening for CKD. The algorithms demonstrate proficiency in identifying microvascular damage and other retinal changes indicative of CKD, suggesting a common pathogenic link between renal and ocular health. Table 3 briefly summarizes key aspects of a representative study.

*Sabanayagam et al. (2020)* pioneered the development of a deep learning algorithm that meticulously analyzes retinal images to screen for CKD in a community setting. Their work emphasizes the potential of retinal images as a credible and adjunctive tool for CKD screening, thereby aligning with the focus on retinal vascular changes as indicators of renal health. *Kang et al. (2020)* further extend this narrative by exploring the relationship between retinal fundus images and renal function impairment. Their research is grounded in the historical context of retinal imaging being a diagnostic tool for both ocular diseases and systemic cardiovascular risks. By employing a deep learning model on a substantial dataset of retinal fundus images, they managed to obtain the model's efficacy in diagnosing early renal function impairments, particularly in patients with elevated HbA1c levels. The study also highlighted the connection between retinal vasculature changes and renal function impairment. *Zhang et al. (2021)* contribute to this discourse by developing an AI-based methodology that determines early renal function impairment using retinal fundus images. Their study not only highlights the model's diagnostic precision but also illuminates the complex relationship between vascular aberrations in the retina and CKD. Analyzing these findings makes it clear that combining AI and machine learning with retinal imaging is a promising approach for early CKD detection. The studies collectively advocate for the potential of these technologies in discovering CKD markers through retinal assessments. However, while the potential is significant, the studies also subtly underscore the need for

**Table 2** **Summary of studies investigating the relationship between retinal changes and CKD.** Relationship between retinal pathologies/features and kidney pathologies/biomarkers in CKD.

| Authors (Year) | Retinal imaging modality/technique | Retinal pathologies/features | Kidney pathologies/biomarkers | Direct correlation? Yes | No | Relationship between retinal and kidney |
|---|---|---|---|---|---|---|
| Edwards et al. (2005) | Retinal photography | Retinopathy (microaneurysms, hemorrhages, exudates) | Serum creatinine levels; eGFR | ✓ | | The presence of retinal exudates and hemorrhages showed a significant increase in serum creatinine levels and a decline in eGFR compared to those without retinopathy. |
| | | Arteriolar diameter | Serum creatinine levels; eGFR | | ✓ | Retinal arteriolar abnormalities did not show a significant association with serum creatinine levels and eGFR. |
| Sabanayagam et al. (2009) | Retinal photography | CRAE; CRVE | eGFR | ✓ | | Retinal arteriolar narrowing, a chronic microvascular damage marker from hypertension, correlates with renal arteriolar changes and CKD |
| Baumann et al. (2010) | Retinal photography | AVR | Serum creatinine levels; eGFR | ✓ | | Narrower retinal arterioles in CKD patients indicate extended CKD effects on cerebral microvasculature. |
| Grunwald et al. (2010) | Retinal photography | Ocular fundus pathology; retinopathy (diabetic and/or hypertensive) | eGFR | ✓ | | Lower eGFR and a higher incidence of fundus pathology. |
| Awua-Larbi et al. (2011) | Retinal photography | CRAE, CRVE | Albuminuria (micro- or macroalbuminuria) | ✓ | | Albuminuria was associated with both narrower and wider arteriolar calibers, |
| Yau et al. (2011) | Retinal photography | CRAE; CRVE | ACR | ✓ | | Retinal arterioles narrowing, may reflect small-vessel damage in the kidney, was associated kidney function impairment in whites but lacks clarity on mechanisms and association in other racial/ethnic groups. |
| Sabanayagam et al. (2011) | Retinal photography | Retinal vessel diameters | Serum creatinine levels; eGFR | | ✓ | Retinal arteriolar and venular diameters, including arteriolar narrowing and venular widening, were not associated with the 15-year risk of incident CKD or significant changes in eGFR |

**Table 2** (*continued*)

| Authors (Year) | Retinal imaging modality/technique | Retinal pathologies/features | Kidney pathologies/biomarkers | Direct correlation? | | Relationship between retinal and kidney |
|---|---|---|---|---|---|---|
| | | | | Yes | No | |
| *Ooi et al. (2011)* | Retinal photography | CRAE; CRVE | eGFR | ✓ | | Arteriolar narrowing linked to CKD and hypertension. |
| *Lim et al. (2013)* | Retinal photography | Arteriolar caliber; Vascular Df; arteriovenous nicking; opacification | ACR; eGFR | ✓ | | Narrower arteriolar caliber, smaller fractal dimensions, arteriovenous nicking, and opacification linked to lower eGFR and higher ACR. |
| *McGowan et al. (2015)* | Retinal photography | Vascular caliber; CRAE; CRVE; Df | eGFR | | ✓ | Retinal arteriolar narrowing is significantly associated with hypertension, but there is no significant link between retinal vascular parameters and CKD. |
| *Phan et al. (2016)* | Retinal photography | Arteriolar diameter; Venular caliber | eGFR | | ✓ | No independent associations between CKD and retinal arteriolar or venular caliber |
| *Gu et al. (2016)* | Retinal photography | CRAE; CRVE | eGFR | ✓ | | CRAE narrowing correlates with renal dysfunction, with CRAE below 150 μm indicating early eGFR decline. |
| *Yip et al. (2017)* | Retinal photography | CRAE; CRVE; Df; Tortuosity; Branching angle; Retinopathy | eGFR | ✓ | | Smaller retinal arterioles, larger retinal venules, and presence of retinopathy were associated with incident CKD. |
| *Vadalà et al. (2019)* | OCT/OCTA | Choroidal thickness | Albuminuria; HbA1c Levels | ✓ | | Significant positive correlation found between choroidal thickness and both albuminuria and HbA1c levels. |
| *Sabanayagam et al. (2020)* | Retinal photography | Deep learning algorithm trained to identify CKD signs in retinal photographs (specific features not detailed). | eGFR | ✓ | | Correlation between retinal microvascular signs and CKD. |
| *Kasumovic et al. (2020)* | OCT/OCTA | FAV perimeter; Foveal & radial peripapillary vessel density; Nonflow area; Flow index; Choriocapillary flow | Albuminuria; eGFR | ✓ | | Reduced vessel density and FAZ alterations, reflect kidney damage. |
| *Paterson et al. (2020)* | Retinal photography & OCT/OCTA | Inner retinal thickness (GCL, IPL); Choroidal thickness; CRAE; CRVE; AVR; Df | eGFR | ✓ | | Higher CRVE is associated with a lower odds ratio for CKD, while AVR and Df show significant associations with CKD stages. |

**Table 2** (*continued*)

| Authors (Year) | Retinal imaging modality/technique | Retinal pathologies/features | Kidney pathologies/biomarkers | Direct correlation? | | Relationship between retinal and kidney |
|---|---|---|---|---|---|---|
| | | | | **Yes** | **No** | |
| *O'Neill et al. (2020)* | Retinal photography | CRAE; CRVE; AVR; Df; Tortuosity | eGFR | ✓ | | Increased retinal venular tortuosity correlates with CKD stages 3–5 (eGFR <60 mL/min/1.73 m$^2$), independent of confounding factors. |
| *Kang et al. (2020)* | Retinal photography | Retinal fundus images | eGFR | ✓ | | A deep learning model using retinal fundus images achieved an AUC of 0.81 in detecting impaired renal function. |
| *Peng et al. (2020)* | OCT/OCTA | Vessel density | eGFR | | ✓ | Long-term systolic blood pressure (SBP) variability is negatively correlated with SVP vessel density. |
| | | FAZ zone size; flow void area in the superficial vascular plexus (SVP) & deep vascular plexus (DVP) | eGFR | ✓ | | Higher average SBP, maximum SBP, within-patient SBP standard deviation, and proportion of high SBP measurements are linked to flow void areas in the SVP, indicating microvascular injury. |
| *Frost et al. (2021)* | OCT/OCTA | Capillary density in the whole image; Fovea; Parafovea | albumin/creatinine ratio; eGFR | ✓ | | Parafoveal retinal capillary rarefaction associated with increased pulse wave velocity, higher log-albumin/creatinine ratio, and reduced eGFR, suggesting a connection between retinal and kidney damage in hypertension |
| *Paterson et al. (2021)* | Retinal photography | Df | Albuminuria | ✓ | | The lower fractal dimension in retinal microvasculature linked to higher odds of albuminuria. |
| | | Tortuosity; Arteriolar; Venular caliber | Albuminuria | | ✓ | No significant association with tortuosity or arteriolar/venular caliber. |
| *Peng et al. (2021)* | OCT/OCTA | Superficial and deep vascular plexus vessel density (DVP-VD); | eGFR | ✓ | | DVP-VD is associated with early cognitive impairment in CKD. |
| | | Retinal nerve fiber layer thickness (RNFLt); Ganglion cell complex (GCC) thickness and loss volumes; FAZ size | eGFR | | ✓ | Nonsignificant association with early cognitive impairment in CKD. |

**Table 2** (*continued*)

| Authors (Year) | Retinal imaging modality/technique | Retinal pathologies/features | Kidney pathologies/biomarkers | Direct correlation? | | Relationship between retinal and kidney |
|---|---|---|---|---|---|---|
| | | | | Yes | No | |
| *Yong et al. (2022)* | OCT/OCTA | FAZ; Vascular density (VD); Perfusion density (PD); Macular volume (MV) | | ✓ | | Significant differences in VD, PD, and MV between control and CKD groups, and among different CKD causes. |
| | | CRVE; Vessel tortuosity indices | eGFR; Proteinuria | ✓ | | |
| *Mustafar et al. (2023)* | Retinal photography, OCT/OCTA | CRAE | eGFR; Proteinuria | | ✓ | eGFR value affects retinal vessel tortuosity and CRVE |

**Notes.**

ACR, Albumin-creatinine ratio; eGFR, estimated Glomerular Filtration Rate; AUC, Area under the ROC Curve; FAZ, Foveal avascular zone; AVR, Arteriole-to-venule ratio; HbA1c, Glycated Hemoglobin; CRAE, Central Retinal Arteriolar Equivalent; OCT, Optical coherence tomography; CRVE, Central Retinal Venular Equivalent; OCT-A, Optical coherence tomography angiography; Df, Fractal dimension.

continued research, validation, and refinement of these AI-driven diagnostic capabilities across diverse and expansive datasets to ensure broader applicability and precision.

Digital fundus images are the focus in these studies as they allow for a detailed examination of retinal vasculature. AI models trained on these images can recognize patterns and features such as retinal vasculature, hemorrhages, and exudations, indicative of renal function impairment. The saliency maps generated from the retinal fundus images have been instrumental in identifying these significant features, thereby connecting the association between retinal and renal health. Therefore, these studies are evident to the promising potential of AI linked with retinal microvasculature changes observed in fundus images as a powerful tool for detecting renal health state. The findings highly suggest for the integration of AI-driven methodologies in screening assessment, emphasizing their potential as early-warning systems for timely interventions in CKD progression. While the results are encouraging, there is a subtle emphasis on the need for continued research to further validate and optimize these AI-driven approaches. Figure 4 show illustrates the heatmaps from a deep learning model analysis of retinal photographs, indicating areas of interest that differentiate between individuals with and without CKD. These visualizations provide insights into the predictive markers of CKD as identified by the AI system, with varying degrees of retinal changes corresponding to the presence and severity of CKD, and associated conditions such as hypertension and DR.

## CHALLENGES AND WAY FORWARD

The complex relationship between retinal pathologies or features and kidney pathologies or biomarkers, particularly in the context of CKD, highlights a promising yet challenging frontier in medical diagnostics. The utilization of retinal photography and OCT/OCTA

**Table 3  Potential of AI and retinal microvasculature changes in CKD detection.**

| Authors (year) | Data source/ sample size | Objective | AI algorithm/ model used | Retinal imaging modality/ technique | Validation strategy | Performance metrics results | Main findings/ results | Limitations |
|---|---|---|---|---|---|---|---|---|
| *Sabanayagam et al. (2020)* | 2,970 retinal images from 6,485 participants in the SEED study. | CKD prediction | DLA using three models: image DLA, risk factors DLA, and hybrid DLA. | Retinal photography | Internal validation with SEED data; External validation on 3,735 patients from the SP2 and 1,538 patients from the BES. | AUC of 0.733–0.938 | Demonstrated good AUC performance, especially in the hybrid model which combines retinal images with clinical risk factors. | Focused on Asian populations; may need validation in other groups. |
| | | | | | | AUC: Image DLA: SEED—0.911 (95% CI [0.886–0.936]), SP2—0.733 (95% CI [0.696–0.770]), BES—0.835 (95% CI [0.767–0.903]); RF DLA: SEED—0.916 (95% CI [0.891–0.941]), SP2—0.829 (95% CI [0.797–0.861]), BES—0.887 (95% CI [0.828–0.946]); Hybrid DLA: SEED—0.938 (95% CI [0.917–0.959]), SP2—0.810 (95% CI [0.776–0.844]), BES—0.858 (95% CI [0.794–0.922]). | | |
| *Kang et al. (2020)* | 25,706 retinal images were obtained from 6,212 patients at Chang Gung Memorial Hospital, Taiwan. | Early renal function impairment prediction | CNN with the VGG-19 architecture | Retinal photography | Non-overlapping training, validation, and testing sets in an 8:1:1 ratio | AUC of 0.81 | The model was more accurate for patients with elevated serum HbA1c levels. | Limited to early renal function impairment; challenges with image quality; higher accuracy in patients with elevated HbA1c, indicating potential variability in performance based on patient condition. |
| | | | | | | HbA1c $\leq$ 6.5%: Sensitivity: 0.84 | | |
| | | | | | | Specificity: 0.62 | | |
| | | | | | | PPV: 0.77 Accuracy: 0.75 | | |
| | | | | | | HbA1c >6.5%: Sensitivity: 0.89 | | |
| | | | | | | Specificity: 0.61 | | |
| | | | | | | PPV: 0.77 | | |
| | | | | | | Accuracy: 0.77 | | |
| | | | | | | HbA1c >7.5%: Sensitivity: 0.89 | | |
| | | | | | | Specificity: 0.60 | | |
| | | | | | | PPV: 0.82 | | |
| | | | | | | Accuracy: 0.79 | | |
**Table 3 (*continued*)**

| Authors (year) | Data source/ sample size | Objective | AI algorithm/ model used | Retinal imaging modality/ technique | Validation strategy | Performance metrics results | Main findings/ results | Limitations |
|---|---|---|---|---|---|---|---|---|
| | | | | | | HbA1c >10.0%: Sensitivity: 0.89 Specificity: 0.61 | | |
| | | | | | | PPV: 0.77 | | |
| | | | | | | Accuracy: 0.77 | | |
| *Zhang et al. (2021)* | 115,344 retinal images from 57,672 patients. | CKD and T2DM prediction | DLA | Retinal photography | Trained on 86,312 images, tested on external cohorts | AUC of 0.93 | High predictive accuracy for CKD and T2DM using fundus images; effective eGFR and glucose level prediction. | Tested primarily on specific populations (T2DM); needs broader validation. |
| *Zhao et al. (2021)* | 418 patients with type 2 diabetes mellitus from Peking Union Medical College Hospital. | Investigate retinal vascular characteristics relationship with renal function in T2DM | NFN + | OCT/OCTA | Statistical analysis | MAE for eGFR prediction ranged from 11.1 to 13.4 mL/min/1.73m$^2$ Correlation coefficients: Df and eGFR: $r = 0.24$, $p < 0.001$ | Showed significant correlations between retinal vascular metrics, particularly Df and various renal function indicators. | Cross-sectional study focused on T2DM patients; may limit generalizability. |
| | | | | | | Df and ACR: $r = -0.21$, $p < 0.001$ | | |
| | | | | | | Df and Cr: $r = -0.15$, $p = 0.003$ | | |
| | | | | | | Df and Alb: $r = 0.17$, $p = 0.001$ | | |

**Notes.**

ACR, Albumin-creatinine ratio; HbA1c, Glycated Hemoglobin; Alb, Albumin; NFN, Neo-fuzzy-neuron; AUC, Area under the ROC Curve; MAE, Mean absolute error; CNN, Convolutional neural network; OCT, Optical coherence tomography; Cr, Creatinine; OCT-A, Optical coherence tomography angiography; Df, Fractal dimension; PPV, Positive Predictive Value; DLA, Deep learning algorithms; RF, Risk Factors; eGFR, estimated Glomerular Filtration Rate; TD2M, Type 2 diabetes.

has proven instrumental in uncovering the connection between retinal features and CKD, emphasizing the necessity for integrated care approaches.

Most of the studies reviewed affirm a direct correlation between retinal and kidney pathologies, indicating that alterations in retinal health often reflect changes in kidney health, especially in diabetic population. This direct correlation is most evident between DR and kidney disease markers such as CKD, microalbuminuria, and eGFR, where the progression of DR is closely associated with the deterioration of kidney function (*Wong et al., 2004a*; *Wong et al., 2004b*; *Klein et al., 2007*; *Pedro et al., 2010*; *Deva et al., 2011*; *Gao et al., 2011*; *Benitez-Aguirre et al., 2012*; *Grunwald et al., 2012*; *Sasongko et al., 2012*; *Liew et al., 2013*; *Nagaoka & Yoshida, 2013*; *Zhang et al., 2014*; *Baumann, Burkhardt & Heemann, 2014*; *Bao et al., 2015*; *Yip et al., 2015*; *Hwang et al., 2016*; *Keel et al., 2017*; *Park et al., 2019*; *Vadalà et al., 2019*; *Zhuang et al., 2020*; *Wang et al., 2020*; *Xu et al., 2020*; *Zhang et al., 2021*; *Nusinovici et al., 2021*; *Zhao et al., 2021*; *Iwase et al., 2023*; *Yan et al., 2023*; *Yazdani et al., 1995*).

The direct correlation between alterations in retinal health, such as changes in CRVE and CRAE, and kidney function markers like eGFR and microalbuminuria, highlights

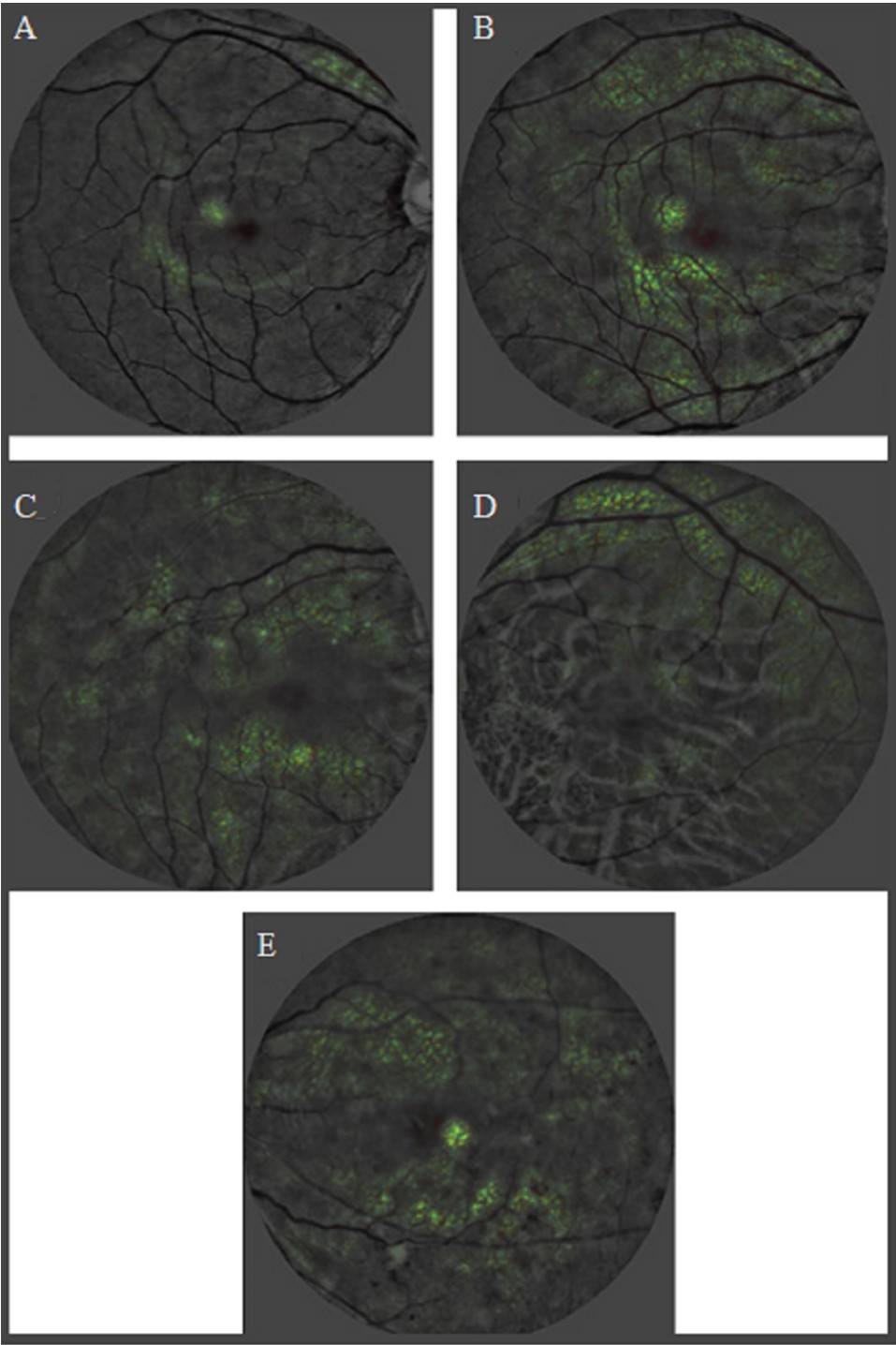

**Figure 4** **Heatmaps of CKD controls and cases adapted from work done by** *Sabanayagam et al. (2020)* **in the attempt to detect CKD from retinal photographs based on deep learning algorithm.** (A) Control: no chronic kidney disease. (B) Control: no CKD. (C) CKD in a person with hypertension. (D) CKD in a person with uncontrolled diabetes. (E) CKD in a person with hypertension, and moderate diabetic retinopathy.

the potential of retinal imaging as a non-invasive, predictive tool for CKD (*Grauslund et al., 2009*; *Sabanayagam et al., 2009*; *Klein et al., 2007*; *Awua-Larbi et al., 2011*; *Ooi et al., 2011*; *Yau et al., 2011*; *Yip et al., 2017*; *O'Neill et al., 2020*; *Paterson et al., 2020*; *Nusinovici et al., 2021*). Additionally, another significant direct correlation between retinal and kidney pathologies is alterations in retinal AVR and specific signs of retinopathy (such as arteriovenous nicking, hemorrhages, exudates, and optic disk edema) being associated with changes in kidney function markers like eGFR and albuminuria (*Wong et al., 2004a*; *Edwards et al., 2005*; *Baumann et al., 2010*; *Pedro et al., 2010*; *Bao et al., 2015*; *Lim et al., 2013*; *O'Neill et al., 2020*; *Paterson et al., 2020*; *Yan et al., 2023*; *Yazdani et al., 1995*).

These connections, while offering a promising avenue for early detection and monitoring of kidney diseases, also introduce complexities in understanding and leveraging these associations effectively. Each of those microvascular changes can reflect varying degrees of microvascular damage or systemic disease progression, making it difficult to establish a one-size-fits-all approach to diagnosis and monitoring. This diversity highlights the complex relationship between retinal and kidney health and necessitates a sophisticated understanding of how each feature correlates with renal pathology.

Compounding this challenge is the task of accurately quantifying these microvascular changes and achieving standardization in their measurement across diverse imaging platforms and clinical settings. The variability inherent in imaging techniques and the subjective nature of image interpretation contribute to inconsistencies in how these microvascular features are assessed. To bridge this gap, there is a pressing need for robust methodologies that can reliably capture and interpret the subtle complexity of retinal microvascular health. Establishing uniform protocols for the quantification and analysis of retinal images is critical for ensuring that assessments of retinal microvascular features can be consistently applied and interpreted, thereby enhancing their reliability as markers for kidney disease progression and facilitating their integration into routine clinical practice.

The future of managing the interplay between retinal pathologies and kidney diseases, particularly related to CKD, lies in utilizing the power of retinal imaging techniques and AI (*Kang et al., 2020*; *Sabanayagam et al., 2020*; *Zhang et al., 2021*; *Zhao et al., 2021*).

The established correlations between retinal changes and CKD highlight the potential for these technologies to revolutionize screening, monitoring, and treatment strategies. Integrating retinal photography (*Kang et al., 2020*; *Sabanayagam et al., 2020*; *Zhang et al., 2021*) and OCT/OCTA (*Zhao et al., 2021*) into routine screenings could lead to the early detection of kidney diseases through the development of predictive models that utilize retinal biomarkers for forecasting kidney function decline. The approach enhanced by advancements in AI and machine learning, promises not only to improve the accuracy and efficiency of early disease detection but also to pave the way for personalized medicine, tailoring interventions to individual risk profiles.

To overcome current challenges and optimize patient management, future research must focus on creating AI models robust across diverse populations to ensure global applicability. Expanding datasets to encompass a wider range of ethnicities and conditions beyond T2DM (*Zhang et al., 2021*; *Zhao et al., 2021*) is essential for improving model accuracy and reliability. Furthermore, enhancing retinal imaging technology to address

image quality issues (*Kang et al., 2020*) and exploring longitudinal studies could provide more dynamic insights (*Zhang et al., 2021*) into CKD's progression. Collaborative efforts across disciplines, including nephrology, ophthalmology, and computational sciences, are crucial for translating these technological advancements into clinical practice, offering a comprehensive care model that addresses the complexity of diabetic complications.

By focusing on these innovative pathways, the medical community can anticipate a significant transformation in CKD detection and management. This multidisciplinary approach not only aims at earlier interventions and improved patient outcomes but also at reducing the global burden of kidney disease, marking a pivotal stride towards a more holistic and effective healthcare paradigm that seamlessly integrates renal and ocular health.

## CONCLUSION

This review has demonstrated that retinal photography, supported by the advancements in OCT and artificial intelligence, has emerged as a critical, non-invasive tool capable of identifying vascular changes indicative of early CKD. Through detailed analysis, we have identified that retinal arteriolar narrowing, specific retinopathy features such as microaneurysms, hemorrhages, and exudates, and changes in CRAE/CRVE are the most consistent indicators linked to the early detection of CKD. These findings highlight the potential of retinal imaging as a transformative diagnostic tool.

However, the relationships involving retinal arteriolar and venular diameters, and certain demographics such as children and adolescents, have shown less consistency. This suggests the need for more targeted studies to confirm their utility in CKD detection across different populations and conditions, highlighting the complexities and variability in diagnosing and monitoring CKD. A major gap identified is the lack of standardized methods to quantify these microvascular changes across diverse imaging platforms and clinical settings, which challenges the consistent application and reliability of retinal imaging as a definitive marker for kidney disease.

To bridge these gaps, future research must focus on establishing uniform protocols for the quantification and analysis of retinal images. Enhancing the reliability of retinal imaging as a diagnostic tool will ensure its consistent application in diverse clinical environments. Moreover, the evolution of AI models and the expansion of data reservoirs are anticipated to significantly increase the precision and predictive capability of these diagnostic tools, enabling more accurate early detection and monitoring of CKD.

As we move forward, collaborative efforts across disciplines—incorporating nephrology, ophthalmology, and computational sciences—are crucial. Such integration will facilitate the translation of technological advancements into clinical practice, offering a comprehensive care model that effectively tackles the complexities of CKD and its complications. Embracing these innovative pathways will lead to significant transformations in CKD management, marking a pivotal stride towards a more holistic and effective healthcare paradigm that seamlessly integrates renal and ocular health.

### Funding

This project was funded by the Research University Grant, Universiti Kebangsaan Malaysia (Grant no. GUP-2022-005). We were supported by Multimedia University throughout the duration of this project. The funders had no role in study design, data collection and analysis, decision to publish, or preparation of the manuscript.

### Grant Disclosures

The following grant information was disclosed by the authors:
Research University Grant, Universiti Kebangsaan Malaysia: GUP-2022-005.

### Competing Interests

The authors declare there are no competing interests.

### Author Contributions

- Nur Asyiqin Amir Hamzah conceived and designed the experiments, performed the experiments, analyzed the data, prepared figures and/or tables, authored or reviewed drafts of the article, and approved the final draft.
- Wan Mimi Diyana Wan Zaki conceived and designed the experiments, performed the experiments, analyzed the data, prepared figures and/or tables, authored or reviewed drafts of the article, and approved the final draft.
- Wan Haslina Wan Abdul Halim conceived and designed the experiments, performed the experiments, authored or reviewed drafts of the article, she is an consultant ophthalmologist at HCTM KL, Malaysia. She provides consultation related to CKD-retinal diseases, and approved the final draft.
- Ruslinda Mustafar conceived and designed the experiments, authored or reviewed drafts of the article, she is a consultant nephrologist and physician at HCTM KL, Malaysia. She provides consultation related to CKD-retinal diseases, and approved the final draft.
- Assyareefah Hudaibah Saad conceived and designed the experiments, performed the experiments, authored or reviewed drafts of the article, and approved the final draft.

### Data Availability

This article is a literature review.

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
