# Peer review of "Evaluating the potential of retinal photography in chronic kidney disease detection: a review"

_PeerJ, doi:10.7717/peerj.17786_

## Round 0.1 · original submission · Major Revisions

Please better explain the selection of the analyzed studies and your aim. Also, please highlight your review novelty and added value.

Reviewer 1 ·

Basic reporting

All comments have been added in detail to the 4th section called additional comments.

Experimental design

All comments have been added in detail to the 4th section called additional comments.

Validity of the findings

All comments have been added in detail to the 4th section called additional comments.

Additional comments

Review Report for PeerJ
(Evaluating the potential of retinal photography in chronic kidney disease detection: A review)

1. Within the scope of the study, both conventional manual evaluations and deep learning approaches in the literature were reviewed in relation to retinal images for the detection of chronic kidney disease.

2. In the introduction part, the definition of chronic kidney disease, its importance, non-invasive condition, eye examination, and its detection with artificial intelligence are mentioned. In addition, the contributions of the study are expressed in four different aspects: tracing methodological evolution, retinal vascular analysis, retinal images in the detection of chronic kidney disease, and the advancement of artificial intelligence.

3. Although review papers examine studies in the literature, their difference from other reviews in the literature on the same subject and therefore their originality must be expressed. For this reason, for the study of chronic kidney disease detection from retinal images discussed here, its difference from other reviews in the literature on the same subject should be explained more clearly.

4. Although important publisher databases are given in the Survey Methodology section, it is recommended to do a quick scan of the last few years of current studies in open access databases such as PeerJ.

5. Normal and abnormal (diabetic retinopathy) retinal images and their contents are given adequately in Figure-2 and relevant explanations are made. Additionally, the literature is clearly expressed in Table-1 depending on the relationship between Retinal Pathologies/Features and Kidney Pathologies/Biomarkers. The explanations and figures in the Evolution of Retinal Vascular Analysis in Chronic Kidney Disease Detection section are generally appropriate.

6. Improvements are needed in the AI and Retinal Microvasculature for Potential CKD Detection section and table. It is recommended to include more recent studies in the literature and to express their advantages and disadvantages more clearly. Only the AUC score is included in the performance metric section; it is recommended to interpret it in terms of other important metrics (such as f1-score).

As a result, although the study is important in terms of summarizing the problem and literature; for further development, it is recommended to include studies in open access databases such as PeerJ, to clearly demonstrate the difference from other reviews in the literature, and to pay attention to the parts mentioned above, such as deepening the AI part.

·

Basic reporting

Revisions needed :-
1. The paper provides a substantial number of references and a solid background context,but one thing I found odd was your reference table split across several pages, with tables appearing in between please fix that.
2. The manuscript generally uses clear and professional English, but frequently I noticed grammatical mistakes and very complicated sentences in the paper, please work on them.
3. Enhance the clarity of the flowchart by including more detailed annotations and ensuring it visually represents all steps mentioned in the text. Make sure it's easily readable and all elements are clearly distinguishable.
4. Simplify the table 1 and provide a clear legend or glossary section that explains abbreviations and terms used. Ensure that the table headings are descriptive and that the table directly relates to the discussed content in the text.
5. Revise the table 2 to clearly differentiate the studies based on outcomes, methodologies, and significance. Each entry should clearly state how it contributes to understanding the link between retinal changes and CKD.
6. The introduction clearly outlines the motivation behind using retinal imaging for CKD detection. However, it could better define the target audience and explicitly state the scientific and clinical implications of the review.

Experimental design

Revisions Needed :-
1. Provide more detailed descriptions of the search terms used, databases accessed, and specific inclusion/exclusion criteria. This will help in replicating the study or understanding the scope of the review.

Comments:-
1. The content fits within the journal’s scope, focusing on innovative diagnostic methods.
2. The manuscript adequately cites sources and the review is well-organized into logical sections.

Validity of the findings

1. The manuscript should better highlight the novelty of its approach and the specific impact it aims to have on the field. Clearly articulate the potential advancements this review brings over previous works.
3. The argument is well-developed but could be strengthened by addressing any new insights or changes in understanding that have emerged since the last reviews.
4. The manuscript does identify some future directions; it could benefit from a clearer delineation of unresolved questions and specific gaps that future research could aim to fill.

---

## Round 0.2 · accepted · Accept

All the previous suggestions were satisfactorily resolved by the authors. The revised version is suitable for publication.

Reviewer 1 ·

Basic reporting

All comments have been added in detail to the last section.

Experimental design

All comments have been added in detail to the last section.

Validity of the findings

All comments have been added in detail to the last section.

Additional comments

Review Report for PeerJ
(Evaluating the potential of retinal photography in chronic kidney disease detection: A review)

Thanks for the revision. The final version of the revised paper and the responses to the reviewer comments have been examined in detail. Due to both the responses and the improvements made to the paper, I recommend that this review paper be accepted as it is. I wish the authors success in their future studies. Kind regards.

·

Basic reporting

1. Clear and unambiguous, professional English used throughout.
No comment. The manuscript is written in clear, professional English.

2. Literature references, sufficient field background/context provided.
No comment. The literature is well-referenced, and the background/context is sufficiently provided.

3. Professional article structure, figures, tables. Raw data shared.
No comment. The article structure conforms to professional standards, and figures and tables are appropriately used.

4. Is the review of broad and cross-disciplinary interest and within the scope of the journal?
Yes, the review is of broad and cross-disciplinary interest and fits well within the journal's scope.

5. Has the field been reviewed recently? If so, is there a good reason for this review (different point of view, accessible to a different audience, etc.)?
The review addresses a timely and relevant topic with a unique perspective on integrating retinal photography and AI for CKD detection.

6. Does the Introduction adequately introduce the subject and make it clear who the audience is/what the motivation is?
Yes, the introduction adequately introduces the subject, clearly stating the motivation and intended audience.

Experimental design

2. Study Design
Article content is within the Aims and Scope of the journal and article type.
No comment.

Rigorous investigation performed to a high technical & ethical standard.
No comment.

Methods described with sufficient detail & information to replicate.
No comment.

Is the Survey Methodology consistent with a comprehensive, unbiased coverage of the subject? If not, what is missing?
No comment.

Are sources adequately cited? Quoted or paraphrased as appropriate?
No comment.

Is the review organized logically into coherent paragraphs/subsections?
No comment.

Validity of the findings

No further comments, my previous suggestions have already been addressed.

Additional comments

The manuscript is well-written and provides a comprehensive review of the potential of retinal imaging in CKD detection. The study's methodology is robust, and the findings are significant. Overall, the manuscript is suitable for publication with minor revisions.